

# Exact solution for two $\delta$-interacting bosons on a ring in the presence of a $\delta$-barrier: Asymmetric Bethe Ansatz for spatially odd states

**Maxim Olshanii[1⋆], Mathias Albert[2,3], Gianni Aupetit-Diallo[2,4], Patrizia Vignolo[2,3] and Steven G. Jackson[5]**

**1** Department of Physics, University of Massachusetts Boston,
Boston Massachusetts 02125, USA
**2** Université Côte d'Azur, CNRS, Institut de Physique de Nice, 06200 Nice, France
**3** Institut Universitaire de France
**4** SISSA, Via Bonomea 265, I-34136 Trieste, Italy
**5** Department of Mathematics, University of Massachusetts Boston,
Boston Massachusetts 02125, USA

⋆ maxim.olchanyi@umb.edu

## Abstract

In this article, we apply the recently proposed Asymmetric Bethe Ansatz method to the problem of two one-dimensional, short-range-interacting bosons on a ring in the presence of a $\delta$-function barrier. Only half of the Hilbert space—namely, the two-body states that are *odd* under point inversion about the position of the barrier—is accessible to this method. The other half is presumably non-integrable. We consider benchmarking the recently proposed $1/g$ expansion about the hard-core boson point [D. Sen, Int. J. Mod. Phys. A 14, 1789 (1999); A. G. Volosniev, D. V. Fedorov, A. S. Jensen, M. Valiente, N. T. Zinner, *Nature Communications* 5, 5300 (2014)] as one application of our results. Additionally, we find that when the $\delta$-barrier is converted to a $\delta$-well with strength equal to that of the particle-particle interaction, the system exhibits the spectrum of its non-interacting counterpart while its eigenstates display features of a strongly interacting system. We discuss this phenomenon in the "Summary and Future Research" section of our paper.

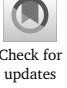
---

# 1  Introduction

The *Bethe Ansatz* is a method for solving a class of many-body problems exactly. Its successes include $\delta$-potential-interacting spinless bosons [1] and spin-1/2 fermions [2,3] on a ring, as well as $N$ $\delta$-potential-interacting bosons in a hard-wall box [3]. At the heart of the Bethe Ansatz solvability there lies a phenomenon of *scattering without diffraction*. The centrality of this phenomenon to Bethe Ansatz is articulated, in particular, in [4]. Here, we focus on $\delta$-function interactions and $\delta$-function barriers, both of which can be regarded as semi-transparent mirrors. Scattering without diffraction manifests as follows: the system's eigenstates consist of reflections of a single plane wave about the constituent mirrors. While this picture is self-evident for a single mirror, two or more mirrors may (and generally will) produce reflected or transmitted plane waves with singular edges, which in turn give rise to a continuum of additional wavevectors. The absence of such discontinuities defines *scattering without diffraction*. The essence of both the conventional Bethe Ansatz and its *Asymmetric Bethe Ansatz* extension [5], which this work relies on, lies in searching for eigenstates as superpositions of plane waves linked by reflections at the interaction mirrors.

The book [6] provides a framework for identifying mirror systems that produce scattering without diffraction and are, consequently, solvable using the Bethe Ansatz. It asserts that such mirrors must form a *generalized kaleidoscope*, i.e., a system of $\delta$-function mirrors invariant (including their coupling constants) under reflection about each constituent mirror. The classification of self-invariant mirror systems without coupling constants is equivalent to listing the so-called *reflection groups*, which are fully tabulated in the mathematics literature [7,8]. A necessary but not sufficient condition for a collection of mirrors to be self-invariant is that the dihedral angles between them must be of the form $\pi/n$, where $n$ is a positive integer. Rules for assigning coupling constants to interaction mirrors appear in limited form in [9] and are fully codified in [5]: *for any two mirrors intersecting at an angle $\pi/n$ with $n$ odd, their coupling constants must be equal*.

The Asymmetric Bethe Ansatz [5] enables exact solutions for mirror systems that violate this rule and, consequently, lack symmetry with respect to a reflection group. Specifically, it allows some coupling constants in a conventionally solvable Bethe Ansatz mirror system to be replaced by hard walls: the walls must correspond to mirrors of a subgroup of the original reflection group. These subgroups are listed and classified in [10]. The Asymmetric Bethe Ansatz has yielded exact eigenstates for two $\delta$-interacting particles in a box with a mass ratio of $1:3$ [5, 11]. Furthermore, it has provided bound states and dimer-barrier scattering states for two $\delta$-interacting bosons in the presence of a $\delta$-barrier [5,12,13], in the sector of the Hilbert space comprising states that are odd under point inversion about the origin. In this article, we focus on the bulk of eigenstates in this sector, i.e., origin-inversion-odd states describing two monomers not trapped by the $\delta$-potential. We assume positive coupling constants for both particle-particle and particle-barrier interactions and apply periodic boundary conditions, in contrast to the hard-wall boundaries considered in [5, 13].

## 2 Statement of the problem

Our goal is to find the eigenstates and eigenenergies of two mass $m$ $\delta$-interacting bosons,

$$\Psi(x_2, x_1) = \Psi(x_1, x_2), \tag{1}$$

on a ring of a circumference $L$,

$$\begin{aligned}
\Psi(x_1 + L, x_2) &= \Psi(x_1, x_2), \\
\Psi(x_1, x_2 + L) &= \Psi(x_1, x_2),
\end{aligned} \tag{2}$$

subject to the field of a $\delta$-barrier. We are focussing on the part of the Hilbert space spanned by the states that are odd with respect to the point inversion about the barrier:

$$\Psi(-x_1, -x_2) = -\Psi(x_1, x_2). \tag{3}$$

Notice that the bosonic symmetry (1) allows one to replace the point inversion anti-symmetry (3) by the anti-symmetry with respect to reflection about the $x_1 = -x_2$ mirror:

$$\Psi(-x_2, -x_1) = -\Psi(x_1, x_2). \tag{4}$$

The Hamiltonian of our system reads

$$\hat{H} = -\frac{\hbar^2}{2m}\frac{\partial^2}{\partial x_1^2} - \frac{\hbar^2}{2m}\frac{\partial^2}{\partial x_2^2} + g\delta(x_1 - x_2) + g_B\delta(x_1) + g_B\delta(x_2). \tag{5}$$

Here and below, $g$ and $g_B$ are the coupling constants responsible for the particle-particle and particle-barrier interactions respectively. We assume both constants be positive:

$$\begin{aligned}
g &> 0, \\
g_B &> 0.
\end{aligned}$$

## 3 Method of solving: Asymmetric Bethe Ansatz

Consider an auxiliary Hamiltonian

$$\hat{H}_{\tilde{C}_2} = -\frac{\hbar^2}{2m}\frac{\partial^2}{\partial x_1^2} - \frac{\hbar^2}{2m}\frac{\partial^2}{\partial x_2^2} + g\delta(x_1 - x_2) + g\delta(x_1 + x_2) + g_B\delta(x_1) + g_B\delta(x_2), \tag{6}$$

subject to the periodic boundary conditions (2). Notice that we added an unphysical, nonlocal interaction $g\delta(x_1 + x_2)$.

According to [6] and, more specifically, [3], the Hamiltonian (6) is Bethe Ansatz solvable. The underlying reflection group is the symmetry group of the tiling of a plane by squares, the group $\tilde{C}_2$ ( [7,8]) generated by the reflections about the following three mirrors (see Fig. 1 for the geometry of the problem):

$$\begin{aligned}
x_1 &= x_2, \\
x_1 &= 0, \\
x_2 &= \frac{L}{2}.
\end{aligned} \tag{7}$$

In the context of the Hamiltonian of interest (5), the Asymmetric Bethe Ansatz recipe [5] dictates the following. One will need to find all the eigenstates of (6) that possess the property (4). What will result is a set of eigenstates that the Hamiltonian (5) shares with (6). Indeed the eigenstates of (6) that are odd with respect to the $x_1 = -x_2$ mirror reflection exhibit neither a discontinuity of the wavefunction nor a discontinuity of its derivatives and as such, do not feel the presence of the $g\delta(x_1 + x_2)$ interaction at all.

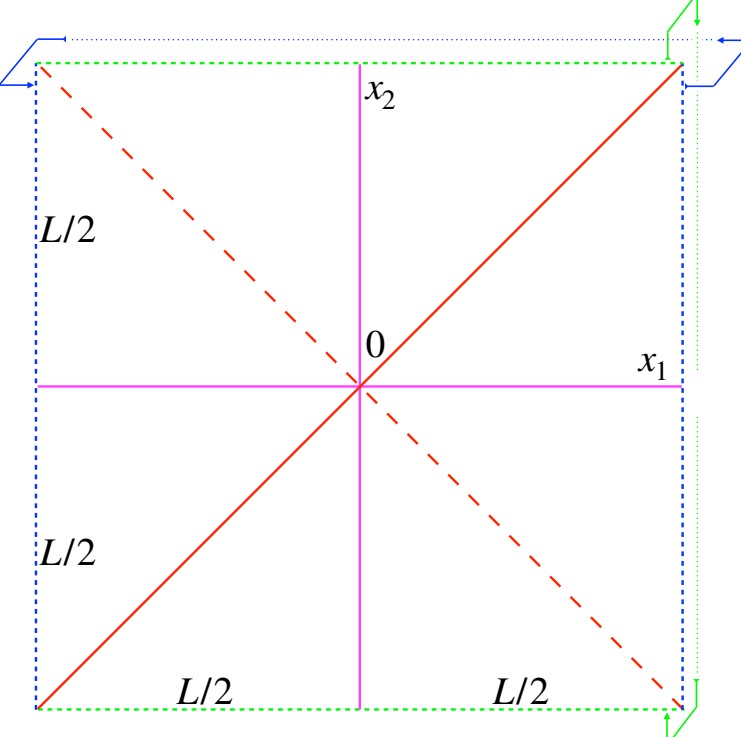

Figure 1: **Geometry of the problem.** According to the Hamiltonian (6), two bosons in a box of side length $L$ with periodic boundary conditions for particle 1 (short-dashed blue) and particle 2 (short-dashed green) interact with a barrier (solid magenta) at the origin. The particles also interact between themselves via a $\delta$-potential peaked at their point of contact (solid red). Additionally, an empirically irrelevant non-local $\delta$-potential is peaked at loci where the particles have equal-magnitude, opposite-sign coordinates (dashed red). This system represents an instance of Gaudin's generalized kaleidoscope [6], solvable via the Bethe Ansatz. The underlying reflection group is the symmetry of a square tiling of a plane, $\tilde{\mathcal{C}}_2$, with generating mirrors at $x_1 = x_2$, $x_1 = 0$, and $x_2 = \frac{L}{2}$.

The empirically relevant Hamiltonian (5) omits the unphysical interaction at $x_1 = -x_2$ (dashed red) and is not generally solvable with the Bethe Ansatz. However, following the Asymmetric Bethe Ansatz recipe [5], the eigenstates of the physical problem that are odd under point inversion about the barrier can be obtained, as they are shared with the unphysical problem. For bosonic particles, the states that are odd under point inversion are also odd with respect to reflection about the $x_1 = -x_2$ mirror featuring a node there as the result.

## 4 Finding the eigenstates and eigenenergies

Since the eigenstates of the empirically relevant Hamiltonian (5) that we are looking for are a subset of the eigenstates of the unphysical but Bethe Ansatz solvable Hamiltonian (6), we will assume that in between the particle-paticle and particle-barrier interaction hyperplanes, $x_1 = x_2$, $x_1 = 0$, and $x_2 = 0$, each eigenstate is represented by a Bethe Ansatz decomposition over all the eight copies (with the wavevectors $\left(\lambda_{1,i}, \lambda_{2,i}\right)$ with $i = 1, \ldots, 8$) of a single two-dimensional plane wave (with a wavevector $(k_1, k_2)$ $(0 < k_1 < k_2)$), under all the eight actions

of the finite subgroup $\mathcal{C}_2$ of the full group of interest, $\tilde{\mathcal{C}}_2$:

$$\left\{ e^{i\left(\lambda_{1,i}x_1 + \lambda_{2,i}x_2\right)} \mid i = 1, \ldots, 8 \right\} \quad \text{with}$$

$$\left\{\left(\lambda_{1,i}, \lambda_{2,i}\right) \mid i = 1, \ldots, 8\right\} = \{(k_1, k_2), (k_2, k_1), (-k_1, k_2), (-k_2, k_1), (k_1, -k_2),$$
$$(k_2, -k_1), (-k_1, -k_2), (-k_2, -k_1)\} \,.$$

We will divide the coordinate space onto six areas,

$$\left\{\mathcal{A}_j \mid j = 1, \ldots, 6\right\} = \left\{0 < x_1 < x_2 \le \frac{L}{2}, \ 0 < x_2 < x_1 \le \frac{L}{2}, \ -\frac{L}{2} < x_1 < x_2 < 0,\right.$$
$$\left.-\frac{L}{2} < x_2 < x_2 < 0, -\frac{L}{2} < x_1 < 0 < x_2 \le \frac{L}{2}, \ -\frac{L}{2} < x_2 < 0 < x_1 \le \frac{L}{2}\right\} \,.$$

Accordingly, we will define six functions of the two coordinates, $\Psi^{(j)}(x_1, x_2)$ with $j = 1, \ldots, 6$, that will govern the behavior of the $(k_1, k_2)$ eigenstates in each of the six areas:

$$\Psi_{k_1, k_2}(x_1, x_2) = \left\{\ \Psi^{(j)}_{k_1, k_2}(x_1, x_2) \quad \text{if} \quad (x_1, x_2) \in \mathcal{A}_j \ \right\} \,.$$

Next, we select two areas, $\mathcal{A}_1$ and $\mathcal{A}_5$, that will serve as the "seeds" informing the rest of the space. Indeed, the symmetries of the problem, i.e. the properties (1) and (4), allow one to reconstruct the wavefunction in the remaining coordinate sectors:

$$\Psi_{k_1, k_2}(x_1, x_2) = \begin{cases} \Psi^{(1)}(x_1, x_2), & \text{for} \quad (x_1, x_2) \in \mathcal{A}_1 \,, \\ \Psi^{(1)}(x_2, x_1), & \text{for} \quad (x_1, x_2) \in \mathcal{A}_2 \,, \\ -\Psi^{(1)}_{k_1, k_2}(-x_2, -x_1), & \text{for} \quad (x_1, x_2) \in \mathcal{A}_3 \,, \\ -\Psi^{(1)}_{k_1, k_2}(-x_1, -x_2), & \text{for} \quad (x_1, x_2) \in \mathcal{A}_4 \,, \\ \Psi^{(5)}_{k_1, k_2}(x_1, x_2), & \text{for} \quad (x_1, x_2) \in \mathcal{A}_5 \,, \\ \Psi^{(5)}_{k_1, k_2}(x_2, x_1), & \text{for} \quad (x_1, x_2) \in \mathcal{A}_5 \,. \end{cases} \tag{8}$$

The wavefuction in the two "seed" domains can be parametrized by two sets of eight coefficients each, $\left(\Psi^{(1)}_{k_1, k_2}\right)_i$ and $\left(\Psi^{(5)}_{k_1, k_2}\right)_i$ with $i = 1, \ldots, 8$:

$$\Psi^{(1)}_{k_1, k_2}(x_1, x_2) = \sum_{i=1}^{8} \left(\Psi^{(1)}_{k_1, k_2}\right)_i e^{i\left(\lambda_{1,i}x_1 + \lambda_{2,i}x_2\right)} \,,$$
$$\Psi^{(5)}_{k_1, k_2}(x_1, x_2) = \sum_{i=1}^{8} \left(\Psi^{(5)}_{k_1, k_2}\right)_i e^{i\left(\lambda_{1,i}x_1 + \lambda_{2,i}x_2\right)} \,. \tag{9}$$

Next, we apply the "jump conditions" induced by the three $\delta$ potentials:

$$\lim_{x_{12} \to 0+} \frac{\partial}{\partial x_{12}} \Psi_{k_1, k_2}\left(X_{12} + \frac{x_{12}}{2}, X_{12} - \frac{x_{12}}{2}\right) - \lim_{x_{12} \to 0-} \frac{\partial}{\partial x_{12}} \Psi_{k_1, k_2}\left(X_{12} + \frac{x_{12}}{2}, X_{12} - \frac{x_{12}}{2}\right) = -\frac{1}{a} \Psi_{k_1, k_2}(X_{12}, X_{12}),$$

$$\lim_{x_1 \to 0+} \frac{\partial}{\partial x_1} \Psi_{k_1, k_2}(x_1, x_2) - \lim_{x_1 \to 0-} \frac{\partial}{\partial x_1} \Psi_{k_1, k_2}(x_1, x_2) = -\frac{1}{a_{\mathrm{B}}} \Psi_{k_1, k_2}(0, x_2),$$

$$\lim_{x_2 \to 0+} \frac{\partial}{\partial x_1} \Psi_{k_1, k_2}(x_1, x_2) - \lim_{x_2 \to 0-} \frac{\partial}{\partial x_1} \Psi_{k_1, k_2}(x_1, x_2) = -\frac{1}{a_{\mathrm{B}}} \Psi_{k_1, k_2}(x_1, 0), \tag{10}$$

where

$$a \equiv -\frac{\hbar^2}{\mu g} \,,$$
$$a_{\mathrm{B}} \equiv -\frac{\hbar^2}{m g_{\mathrm{B}}} \,, \tag{11}$$

are the particle-particle and particle-barrier scattering lengths respectively,

$$\mu = m/2,$$

being the reduced mass. We obtain two linearly independent sets of the sixteen unknown coefficients (9), $\left(\Psi_{k_1,k_2}^{0<x_1<x_2}\right)_i$ and $\left(\Psi_{k_1,k_2}^{x_1<0<x_2}\right)_i$, as a result.

(Note that while the scattering length $a$ governs the relative motion of two particles, the scattering length $a_B$ is responsible for the cartesian motion of a single particle. Hence, the reduced mass $\mu$ in the former case and the true mass $m$ in the latter case. In both cases, the corresponding scattering length is the position of the first node of the even scattering wave in the limit of zero energy.)

Finally, we apply the periodic boundary conditions (2). As a result, we (a) find a unique linear combination of the two linearly independent free-space solutions from the previous step and (b) identify the allowed values of the seed wavevector components $k_1$ and $k_2$. The latter constitute the so-called Bethe Ansatz Equations for our problem.

The resulting (un-normalized) eigenstates are defined by

$$
\begin{aligned}
\Psi^{(1)}(x_1, x_2) = &-(k_1^2 - k_2^2)a^2 \sin[(k_1 L)/2] && (12)\\
&\times \cos[1/2(k_1 + k_2)(x_1 - x_2)]\cos[1/2(k_1 - k_2)(x_1 + x_2)]\\
&+(k_1^2 - k_2^2)a^2 \sin[(k_1 L)/2]\\
&\times \cos[1/2(k_1 - k_2)(x_1 - x_2)]\cos[1/2(k_1 + k_2)(x_1 + x_2)]\\
&+2(k_1 + k_2)a \sin[(k_1 L)/2]\\
&\times \cos[1/2(k_1 + k_2)(x_1 + x_2)]\sin[1/2(k_1 - k_2)(x_1 - x_2)]\\
&-2(k_1 - k_2)a \sin[(k_1 L)/2]\\
&\times \cos[1/2(k_1 - k_2)(x_1 + x_2)]\sin[1/2(k_1 + k_2)(x_1 - x_2)]\\
&-(k_1 + k_2)a(2k_1 a_B \cos[(k_1 L)/2] + (-2 + k_1^2 a a_B - k_1 k_2 a a_B)\sin[(k_1 L)/2])\\
&\times \cos[1/2(k_1 + k_2)(x_1 - x_2)]\sin[1/2(k_1 - k_2)(x_1 + x_2)]\\
&+(-4k_1 a_B \cos[(k_1 L)/2] + 2(2 - k_1^2 a a_B + k_1 k_2 a a_B)\sin[(k_1 L)/2])\\
&\times \sin[1/2(k_1 + k_2)(x_1 - x_2)]\sin[1/2(k_1 - k_2)(x_1 + x_2)]\\
&-(k_2 - k_1)a(2k_1 a_B \cos[(k_1 L)/2] + (-2 + k_1^2 a a_B + k_1 k_2 a a_B)\sin[(k_1 L)/2])\\
&\times \cos[1/2(k_1 - k_2)(x_1 - x_2)]\sin[1/2(k_1 + k_2)(x_1 + x_2)]\\
&+(4k_1 a_B \cos[(k_1 L)/2] + 2(-2 + k_1^2 a a_B + k_1 k_2 a a_B)\sin[(k_1 L)/2])\\
&\times \sin[1/2(k_1 - k_2)(x_1 - x_2)]\sin[1/2(k_1 + k_2)(x_1 + x_2)],\\
\Psi_{k_1,k_2}^{(5)}(x_1, x_2) = &-k_1(k_1 + k_2)a a_B(2\cos[(k_1 L)/2] + (k_1 - k_2)a \sin[(k_1 L)/2]) && (13)\\
&\times \cos[1/2(k_1 + k_2)(x_1 - x_2)]\sin[1/2(k_1 - k_2)(x_1 + x_2)]\\
&+(-4k_1(a + a_B)\cos[(k_1 L)/2] + (4 + k_2^2 a^2 + 2k_1 k_2 a a_B - k_1^2 a(a + 2a_B))\\
&\times \sin[(k_1 L)/2])\sin[1/2(k_1 + k_2)(x_1 - x_2)]\sin[1/2(k_1 - k_2)(x_1 + x_2)]\\
&+k_1(k_1 - k_2)a a_B(2\cos[(k_1 L)/2] + (k_1 + k_2)a \sin[(k_1 L)/2])\\
&\times \cos[1/2(k_1 - k_2)(x_1 - x_2)]\sin[1/2(k_1 + k_2)(x_1 + x_2)]\\
&+(4k_1(a + a_B)\cos[(k_1 L)/2] + (-4 - k_2^2 a^2 + 2k_1 k_2 a a_B + k_1^2 a(a + 2a_B))\\
&\times \sin[(k_1 L)/2])\sin[1/2(k_1 - k_2)(x_1 - x_2)]\sin[1/2(k_1 + k_2)(x_1 + x_2)],
\end{aligned}
$$

with the rest of the coordinate space being restored through (8).

The Bethe Ansatz Equations for rapidities read

$$\cos\left(\frac{k_1 L}{2}\right)\left(\left(k_2^2 a a_\mathrm{B} - 2\right)\sin\left(\frac{k_2 L}{2}\right) + k_2(a + 2a_\mathrm{B})\cos\left(\frac{k_2 L}{2}\right)\right)$$
$$+ k_1 a \sin\left(\frac{k_1 L}{2}\right)\left(k_2 a_\mathrm{B}\cos\left(\frac{k_2 L}{2}\right) - \sin\left(\frac{k_2 L}{2}\right)\right) = 0, \tag{14}$$

$$k_1^2 a \sin\left(\frac{k_1 L}{2}\right)\sin\left(\frac{k_2 L}{2}\right) + 2k_1 \sin\left(\frac{k_2 L}{2}\right)\cos\left(\frac{k_1 L}{2}\right)$$
$$= k_2^2 a \sin\left(\frac{k_1 L}{2}\right)\sin\left(\frac{k_2 L}{2}\right) + 2k_2 \sin\left(\frac{k_1 L}{2}\right)\cos\left(\frac{k_2 L}{2}\right), \tag{15}$$

where $\quad 0 < k_1 < k_2$.

For given set of rapidities, the energy of the state is

$$E_{k_1, k_2} = \frac{\hbar^2}{2m}(k_1^2 + k_2^2). \tag{16}$$

Figure 2 outlines the process of finding the rapidities and the corresponding eigenstates, for a typical set of parameters.

Two remarks are in order:

- The way the Bethe Ansatz Equations are presented does not respect the bosonic symmetry manifestly. The reason is as follows. While the equations (14) and (15) can indeed be combined into a $1 \leftrightarrow 2$ version of the equation (14), the resulting equation will become identical to (14) in the no-barrier limit $a_\mathrm{B} \to \infty$.

- Unlike in the Lieb-Liniger case [1], it is not evident if there exists a universal way of indexing of the solutions of (14) and (15) with two integers.

## 5 Testing the $1/g$ expansions

In [14, 15], it was suggested that in the limit of infinitely strong particle-particle interactions, $g \to +\infty$, the $1/g$ correction to the energy of the resulting hard-core gas can be obtained by formally applying the first order perturbation theory formulas with an effective $1/g$ "operator" as a perturbation. In the context of our problem, the perturbation operator of [14, 15] is represented by

$$\hat{V} = \tilde{g}\,\overleftrightarrow{\partial}_{x_1 - x_2}\,\delta(x_1 - x_2)\,\overrightarrow{\partial}_{x_1 - x_2},$$
$$\tilde{g} = -\frac{\hbar^4}{\mu^2 g}, \tag{17}$$

as a perturbation. Here, the derivative $\overleftrightarrow{\partial}\,(\overrightarrow{\partial})$ acts on the bra(ket) of the corresponding matrix element. Note that here, the analogy with the perturbation theory is superficial, since the "operator" (17) can not be used beyond the first order.[1] Below, we will be able to illustrate this statement by showing that the $1/g^2$ correction to the ground state energy of our system is positive, in contradiction to a general perturbation theory result [16] (§38 there).

---

[1]See Problem 5.1.11 in [20].

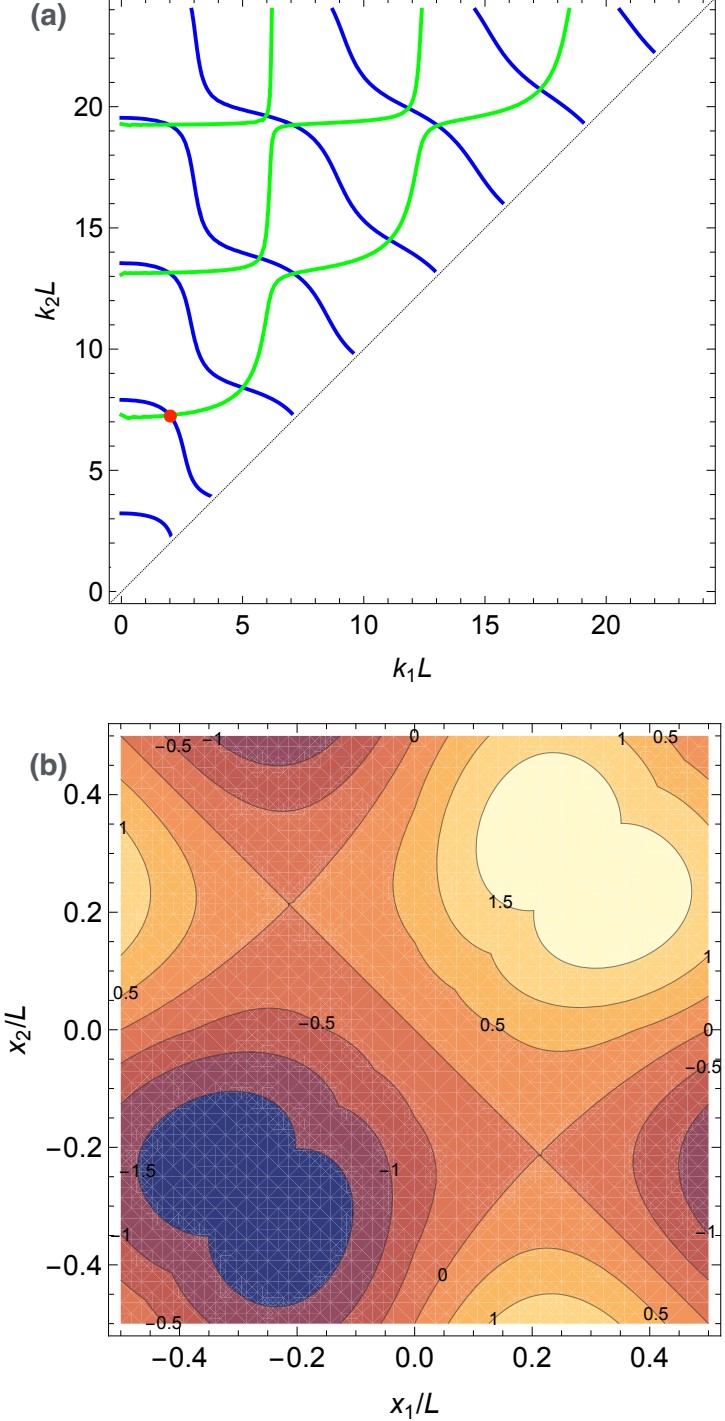

Figure 2: **Finding the spatially odd eigenstates for two $\delta$-interacting bosons interacting with a $\delta$-barrier, subject to periodic boundary conditions.** (a) Solution manifolds for the Bethe Ansatz Equation (14) (blue) and (15) (green). Their intersections provide one with the allowed pairs of rapidities, $(k_1, k_2)$. The red dot marks the ground state. The parameters are $g = 4. \times \hbar^2/mL$ and $g_{\rm B} = g/\sqrt{2}$. (b) The ground state wavefunction. It's rapidities are $k_1 = 2.00 \times 1/L$ and $k_2 = 7.27 \times 1/L$, leading to the ground state energy of $56.7 \times \hbar^2/mL^2$. The wavefunction is normalized to unity.

The Asymmetric Bethe Ansatz $1/g$ expansion is

$$E(k_1, k_2) = \frac{\hbar^2}{mL^2}\left\{\varepsilon(\eta_1, \eta_2) + \varepsilon(\eta_2, \eta_1) + \mathcal{O}(\frac{1}{\xi^3})\right\}, \quad \text{where}$$

$$\varepsilon(\eta_\mathrm{I}, \eta_\mathrm{II}) = \frac{1}{2}\eta_\mathrm{I}^2 - \frac{2\eta_\mathrm{I}^2(\eta_\mathrm{I}^2 + \xi_\mathrm{B}^2)}{\eta_\mathrm{I}^2 + \xi_\mathrm{B}(2 + \xi_\mathrm{B})} \times \frac{1}{\xi} +$$

$$\times \frac{2\eta_\mathrm{I}^2(\eta_\mathrm{I}^2 + \xi_\mathrm{B}^2)(\eta_\mathrm{I}^4(\eta_\mathrm{II}^2 + 3\xi_\mathrm{B}) + \xi_\mathrm{B}^2(2 + \xi_\mathrm{B})(3\xi_\mathrm{B}^2 + \eta_\mathrm{II}^2(2 + \xi_\mathrm{B})) + 2\eta_\mathrm{I}^2\xi_\mathrm{B}(\eta_\mathrm{II}^2(2 + \xi_\mathrm{B}) + \xi_\mathrm{B}(5 + 3\xi_\mathrm{B})))}{\xi_B(\eta_\mathrm{I}^2 + \xi_\mathrm{B}(2 + \xi_\mathrm{B}))^3}$$

$$\times \frac{1}{\xi^2}.$$

$$(18)$$

Here

$$\eta_{1,2} \equiv \kappa_{1,2}L,$$

$$\xi \equiv \frac{g}{\left(\frac{\hbar^2}{mL}\right)},$$

$$\xi_\mathrm{B} \equiv \frac{g_\mathrm{B}}{\left(\frac{\hbar^2}{mL}\right)},$$

where

$$\kappa_{1,2} = \sqrt{2m\epsilon_{1,2}}/\hbar,$$

and $\epsilon_{1,2}$ are two distinct solutions of the one-body Schrödinder equation in presence of the barrier but in the absence of interactions:

$$-\frac{\hbar^2}{2m}\frac{\partial^2}{\partial x^2}\phi(x) + g_\mathrm{B}\delta(x)\phi(x) = \epsilon\phi(x),$$

$$\phi(x + L) = \phi(x).$$

The functions $\phi(x)$ are the fermionic orbitals that are used to build the Tonks-Girardeau wavefunction for hard-core bosons [17], in presence of the barrier [18].

In (18), the $1/g$ term is identical to the one that can be obtained applying the first order perturbation theory to (17). However notice that for repulsive particle-particle and particle-barrier interactions, the $1/g^2$ correction to energy, including the ground state energy (within the point-inversion-odd bosonic sector) is positive, contradicting [16](§38). This indicates that there is no "real world" potential behind the potential-like object (17), and the use of latter must be restricted to the first order perturbation theory.

# 6 Summary and future research

In this paper, we present an exact solution for two one-dimensional $\delta$-interacting bosons subject to periodic boundary conditions in the presence of a $\delta$-function barrier, within the sector of the system's Hilbert space that is odd under point inversion about the barrier. Both the particle-particle interaction and the particle-barrier interaction are repulsive. The set of implicit equations determining the (discrete) spectrum of Bethe rapidities $(k_1, k_2)$—analogous to the standard Bethe Ansatz equations for the Lieb-Liniger model [1]—is given in Eqs. (14)-(15). The energy spectrum, given by Eq. (16), can be directly derived from the rapidity spectrum. The wavefunction for each allowed rapidity pair is provided by Eqs. (8), (12), and (13).

This article focuses exclusively on the case of repulsive interactions and a repulsive barrier. As noted previously, bound states for attractive interactions, an attractive barrier, or both, in the absence of periodic boundary conditions, were identified in [5]. Additionally, the scattering of a dimer on a repulsive barrier is addressed in [13].

Let us stress that unlike in [5] and in [13], in this article we replace an open line with periodic boundary conditions. This allows one to study such properties as the level spacing statistics. Consequently, we regard the current article as the first stage of a broader study of quantum chaos in systems that are not chaotic classically. There the paradigmatic model is the *even* with respect to the point inversion sector of the physical system considered in our paper, with the *odd* sector used as an integrable reference [19]. Note that at the level of the classical trajectories, interactions present in either odd or even sectors cannot alter the set of the absolute values of the momenta involved. However, in the latter case, diffraction can introduce new momenta making the even sector more chaotic than its odd counterpart.

States with complex rapidities (strings) under periodic boundary conditions remain a subject for future investigation. Real rapidity states for attractive interactions and/or an attractive barrier are also of interest, despite being obtainable via analytic continuation from the repulsive case. In this parameter range, one phenomenon stands out. When the barrier strength is $a_{\rm B} = -a/2$ (i.e., $g_{\rm B} = -g$), the Bethe Ansatz equations reveal that the energy spectrum matches that of a non-interacting, barrier-free system. However, the ground state (and likely other eigenstates) can be shown to retain the characteristic discontinuities induced by $\delta$-interactions. We currently lack an explanation for this phenomenon, which awaits further exploration in future research.

# Acknowledgments

We are immeasurably grateful to Vanja Dunjko and Anna Minguzzi for numerous discussions.

**Funding information**  M.O. was supported by the NSF Grant No. PHY-2309271. M.O. is grateful to Université Côte d'Azur for providing support for a one-month invited professor postion. M.A. and P.V. acknowledge financial support from the ANR-21-CE47-0009 Quantum-SOPHA project and from the ANR-23-PETQ-0001 Dyn1D at the title of France 2030.

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
