# Peer review of "Exact Solution for Two $δ$-Interacting Bosons on a Ring in the Presence of a $δ$-Barrier: Asymmetric Bethe Ansatz for Spatially Odd States"

_SciPost Physics Core, doi:SciPost Phys. Core 8, 083 (2025)_

## Round 1 · Referee Report · Anonymous (Referee 1) · 2025-10-5

Strengths
- construction of exact eigenstates of the Hamiltonian of an interesting few-body problem: two repulsively interacting bosons on a ring scattering on a static delta barrier potential
Weaknesses
-
the model and the method is the same as in Refs. [12,13] (although, these references were for attractive bosons), so it is not entirely clear where is the innovation in this new work
-
not all the eigenstates are obtained, only the ones corresponding to an integrable subsector. Not much is said about the eigenstates that are not of this form
-
the writing could be improved: some notations are not clear, some pages of the manuscript look merely like working notes and should still be polished
Report
The paper is interesting and will be useful for experts working on this type of problems. I believe that, after appropriate revision, it can be suitable for Scipost Physics Core. However the present version suffers from several issues that should be adressed first, see list below.
Requested changes
-
As the model and method have been introduced in previous works, e.g. Refs. [12,13], the distinction between what was already known and what is new in this work should be made more explicit.
-
I think the eigenstates of the model that are not of the asymmetric Bethe Ansatz form should at least be discussed. Can anything be said about those? Do we have any idea about how many such eigenstates there are, and where they would be in the spectrum? Perhaps the spectrum could be shown, highlighting the eigenstates that are of the form studied in this paper, and the ones that are not? Or is this the purpose of the strangely positioned figure 2, which appears after the bibliography, and is barely referred to in the text?
-
The notations should be clarified. Formulas from page 4 to page 7 are difficult to read. (For instance: How is $\psi^{0<x_1<x_2}$ defined the first time it appears at the top of page 5?)
More minor points:
(i) I suggest to use conventions where $\hbar = m=1$. Also, please avoid introducing unnecessary notations such as '$\mu = m/2$' (no need for a new notation just for a factor $1/2$). By the way, the second formula (11) is probably wrong and should be (in the right units) '$a_B = -1/g_B$', not '$-1/g$'.
(ii) The use of references is a bit sloppy. The authors refer to 'the book [6]', but then it turns ref. [6] is not a book. Same for ref. [15]
Recommendation
Ask for minor revision

---

## Round 3 · Referee Report · Anonymous (Referee 1) · 2025-10-24

Strengths

same as in previous report

Weaknesses

same as in previous report

Report

The authors have answered my comments satisfactorily (except my comment about references; the authors claim they made a correction but nothing has changed, refs. [6,16] are not references but rather references to references). I think the paper is clear enough, and it can be published as it is.

Recommendation

Publish (meets expectations and criteria for this Journal)

---

## Round 3 · Author Response

We are truly grateful for this opportunity to improve our article.

---

## Round 3 · List of Changes

>> As the model and method have been introduced in previous works, e.g. Refs. [12,13], the distinction between what was already known and what is new in this work >> should be made more explicit.

The second to last paragraph of Sec. 6 addresses this explains why the periodic boundary conditions and the unbounded spectrum are so important. In short, our manuscript is the stage one of a plan to address appearance of quantum chaos in systems that a classically regular.

>> I think the eigenstates of the model that are not of the asymmetric Bethe Ansatz form should at least be discussed.

We address them in the same paragraph (second to last paragraph of Sec. 6).

>> Can anything be said about those? Do we have any idea about how many such eigenstates there are, and where they would be in the spectrum? >> Perhaps the spectrum could be shown, highlighting the eigenstates that are of the form studied in this paper, and the ones that are not? >> Or is this the purpose of the strangely positioned figure 2, which appears after the bibliography, and is barely referred to in the text?

The even sector requires hard numerics. We don't need it for the odd states because of the integrability. Incidentally, such a numerical calculation is the very subject of our current research, almost ready for submission. In the new version, we cite this text under Ref. 20.

>> The notations should be clarified. Formulas from page 4 to page 7 are difficult to read. (For instance: How is ψ0<x1<x2 defined? The first time it appears >> at the top of page 5?)

We changed the notations completely.

>> More minor points:

>> (i) I suggest to use conventions where ℏ=m=1. >> Also, please avoid introducing unnecessary notations such as 'μ=m/2' (no need for a new notation just for a factor 1/2).

==

While the scattering length 'a' governs the relative motion of two particles, the scattering length 'a_B' is responsible for the cartesian motion of a single particle. Hence, the reduced mass μ in the former case and the true mass m in the latter case. In both cases, the corresponding scattering length is the position of the first node of the even scattering wave in the limit of zero energy. For that reason, it is important to keep 'hbar' and 'm'.

>> By the way, the second formula (11) is probably wrong and should be (in the right units) 'aB=−1/gB', not '−1/g

Corrected.

>> (ii) The use of references is a bit sloppy. The authors refer to 'the book [6]', but then it turns ref. [6] is not a book. Same for ref. [15]

Corrected

---

## Editorial Decision

published